# Synthesis, Stability, and Bioavailability of Nicotinamide Riboside Trioleate Chloride

**DOI:** 10.3390/nu14010113

**Published:** 2021-12-27

**Authors:** Amin Zarei, Leila Khazdooz, Sara Madarshahian, Mojtaba Enayati, Imann Mosleh, Tiantian Lin, Bing Yan, Gerhard Ufheil, Timothy James Wooster, Alireza Abbaspourrad

**Affiliations:** 1Department of Food Science, College of Agriculture and Life Sciences, Cornell University, Ithaca, NY 14853, USA; az534@cornell.edu (A.Z.); lk532@cornell.edu (L.K.); sm2643@cornell.edu (S.M.); enayati@cornell.edu (M.E.); imosleh@cornell.edu (I.M.); tl475@cornell.edu (T.L.); 2Nestlé Product Technology Center, Nestlé Health Science, Bridgewater, NJ 08807, USA; bing.yan@rd.nestle.com (B.Y.); gerhard.ufheil@rd.nestle.com (G.U.); 3Institute of Material Science, Nestlé Research, CH-1000 Lausanne, Switzerland; timothyjames.wooster@rdls.nestle.com

**Keywords:** nicotinamide riboside trioleate chloride, nicotinamide riboside, synthesis, stability, bioavailability

## Abstract

Nicotinamide riboside chloride (NRCl) is an effective form of vitamin B3. However, it cannot be used in ready-to-drink (RTD) beverages or high-water activity foods because of its intrinsic instability in water. To address this issue, we synthesized nicotinamide riboside trioleate chloride (NRTOCl) as a new hydrophobic nicotinamide riboside (NR) derivative. Contrary to NRCl, NRTOCl is soluble in an oil phase. The results of stability studies showed that NRTOCl was much more stable than NRCl both in water and in oil-in-water emulsions at 25 °C and 35 °C. Finally, we evaluated the bioavailability of NRTOCl by studying its digestibility in simulated intestinal fluid. The results demonstrated that NRTOCl was partially digestible and released NR in the presence of porcine pancreatin in a simulated intestinal fluid. This study showed that NRTOCl has the potential to be used as an NR derivative in ready-to-drink (RTD) beverages and other foods and supplement applications.

## 1. Introduction 

Nicotinamide adenine dinucleotide (NAD^+^) is a vital coenzyme for redox reactions in cellular energy metabolism and mitochondrial functions [1,2]. Additionally, in non-redox cellular reactions, NAD^+^ serves as a crucial cofactor regulating the activity of two essential protein families, sirtuins (SIRTs) and poly (ADP-ribose) polymerases (PARPs) [3,4,5]. The sirtuins play several key roles in maintaining nuclear, mitochondrial, cytoplasmic, or metabolic homeostasis, while important roles of PARPs are repairing DNA and maintaining chromatin structure and function [3,4,5]. NAD^+^ levels decrease during the aging process and can cause defects in nuclear and mitochondrial functions [6,7,8,9,10]. Therefore, supplementation with NAD^+^ precursors can restore NAD^+^ levels and provide health benefits in many animals and humans [7,11,12,13,14,15,16,17,18,19,20,21,22,23,24]. In lactating women, NAD^+^ supplementation can increase milk production and improve the quality of breastmilk [13]. Recent research has demonstrated that high levels of NAD^+^ can help prevent and treat liver cancer [25]. A new study has shown that supplementing NAD^+^ may boost innate immunity to coronaviruses and restrict viral infection [26].

Traditional sources of NAD^+^ include niacin and nicotinamide. Nicotinamide riboside (NR), a new form of vitamin B3 supplement, has also been shown to be an NAD^+^ precursor that is orally available and can roughly double the level of NAD^+^ in mammalian cells [17,27]. NR is more effective than other NAD^+^ precursors, such as niacin and nicotinamide, because it is metabolized to NAD^+^ in fewer steps [27]. NRCl is an FDA-approved nutritional supplement used to boost the NAD^+^ level and is sold as its chloride salt (NRCl) in a capsulated form under the brand name Niagen^TM^ [28].

Unfortunately, NRCl is easily hydrolyzed in an aqueous solution and is, therefore, unsuitable for incorporation into beverages, and other foods and supplements, with high-water activity. Structurally, NRCl is a quaternary ammonium salt containing a sensitive *N*-glycosidic bond that can spontaneously cleave in an aqueous solution, yielding nicotinamide and D-ribose decomposition products (Figure 1). This means that it will be difficult to develop NRCl for ready-to-drink (RTD) beverages or high-water activity foods or supplements.

To overcome this challenge, we sought to make NRCl more stable in aqueous solutions without losing its usefulness as a precursor to NAD^+^. Therefore, we report the synthesis of nicotinamide riboside trioleate chloride (NRTOCl) as an oil-soluble hydrophobic NR derivative (Figure 2). The synthetic design of NRTOCl was predicated on the idea that the long arms of the oleate esters would provide protection to the *N*-glycosidic bond from hydrolysis in aqueous solutions. Further, we chose chloride as the counterion to be food safe [29]. Details of the synthesis of NRTOCl from NRCl are reported here, as well as the digestion and release of NR and its subsequent enzymatic conversion to NAD^+^.

## 2. Methods

### 2.1. Chemicals and Materials

Nicotinamide riboside chloride (beta form) was a gift from Nestlé. Oleoyl chloride was purchased from Aldrich at 89% purity, silica gel (P60, 40–63 µm, 60 Å) was purchased from SiliCycle and Silica Gel 60 F254 Coated Aluminum-Backed TLC Sheets were purchased from EMD Millipore (Billerica, MA, USA). Bovine bile (B3883) and pancreatin from the porcine pancreas (P7545, 8 × USP) were purchased from Aldrich. Sodium caseinate (InfG LwSp Kshr) and lecithin (Lecithin Soy Fluid 50% AIM Kosher) were food grade from Nestlé Product Technology Center, Bridgewater NJ, USA. Canola oil was food grade, pure, and purchased from the local market. MCT oil was food grade and extracted from pure organic coconut oil. It was purchased from the local market.

### 2.2. Instrumentation

A 500 MHz NMR (Bruker AVANCE) spectrometer was used for ^1^H NMR (500 MHz) and ^13^C NMR (125 MHz) spectra in CDCl_3_. Fourier transform infrared spectra (ATR-FTIR) were recorded on a Shimadzu IRAffinity-1S spectrophotometer. UV–vis was recorded on a Shimadzu UV-2600 spectrophotometer. An Agilent 1200 LC System equipped with Binary SL Pump and Diode Array Detector and a Shodex RI-501 Refractive Index Detector (single channel) was used to perform the high-performance liquid chromatography (HPLC) measurements. The HPLC was equipped with an ultraviolet detector (HPLC-UV). Reversed-phase HPLC was performed on a Luna C18(2) 100 Å (150 mm × 4.6 mm), and the column temperature was set at 25 °C. The injection volume was 10.0 μL and ammonium acetate (20 mM) was used as the mobile phase with a flow rate of 0.7 mL min^−1^ over 45 or 60 min. All samples were passed through a 13 mm nylon syringe filter with a 0.22 μm pore size before measurement prior to injection. For LC-MS analysis, we used an LC (Agilent 1100 series) coupled with a mass spectrometer. Reverse-phase chromatography was used with a Phenomenex Luna Omega (Phenomenex) LC column with the following specifications: 100 × 4.6 mm, 3 µm, polar C18, 100 Å pore size with a flow rate of 0.3 mL min^−1^. LC eluents include MiliQ-water (solvent A) and acetonitrile (solvent B) using gradient elution (solution A:B composition change with time: 0 min: 95:5, 3 min: 95:5, 15 min: 85:15, 17 min: 90:10, and 20 min 95:5). The mass spectrometer (Finnigan LTQ mass spectrometer) was equipped with an electrospray interface (ESI) set in positive electrospray ionization mode for analyzing the NRTOCl, NRCl, and nicotinamide. The optimized parameters were a sheath gas flow rate at 20 arbitrary units, spray voltage set at 4.00 kV, the capillary temperature at 350 °C, the capillary voltage at 41.0 V, and tube lens set at 125.0 V. The particle size distribution, mean particle diameter (zeta average size) and zeta-potential of NRTOCl in DI water were measured using a commercial dynamic light-scattering device (Nano-ZS, Malvern Instruments, Worcestershire, UK). Tecnai F20 TEM/STEM transmission electron microscope (200 kV) was used for characterizing the structure and morphology of prepared NRTOCl nanoparticles.

### 2.3. General Procedure for the Synthesis of Nicotinamide Riboside Trioleate Chloride (NRTOCl)

To a round bottom flask in an ice bath fitted with a magnetic stir bar, septa, and nitrogen inlet, 200 mg (0.690 mmol) of NRCl, 0.550 mL (6.81 mmol) of pyridine, and 4.75 mL of DMF were added. Then, 2.00 mL (5.38 mmol) of oleoyl chloride were added dropwise with stirring. The reaction mixture was stirred in an ice bath for 3 h under a nitrogen blanket. The progress of the reaction was followed by thin-layer chromatography (TLC). After 3 h, 5 mL of methanol was added to the reaction mixture to quench the excess oleoyl chloride. The solvent was removed using a rotary evaporator. The crude product was extracted in hexane and finally purified using column chromatography on silica gel using CH_3_OH (12%) and EtOAc (88%) as eluent. The purified NR–trioleate chloride was obtained in 64.3% (479.2 mg) as a pale-cream-colored greasy product (λ_max_ in methanol was 267 nm).

### 2.4. Preparation of 15 wt% NRTOCl in Canola Oil as a Stock for Making Oil-in-Water Emulsions

To a 15 mL falcon tube containing 380 mg (0.35 mmol) of NRTOCl was added 2154 mg of canola oil. The tube was placed in a water bath at 35 °C and shaken until the NRTOCl was completely dissolved in the canola oil. The sample was then cooled to room temperature and used for the next set of studies. Throughout storage, the NRTOCl in canola oil remained clear and with no precipitation.

### 2.5. Preparation of Aqueous Phase for Making NRTOCl in Oil-in-Water Emulsion Using Na-Caseinate (2 wt%), KCl (0.3 wt%), NaCl (0.1 wt%), CaCl_2_ (0.2 wt%), and NaN_3_ (0.01 wt%)

To a 200 mL beaker containing 100 mL of DI water at 70 °C, 2 g of sodium caseinate was gradually added and the mixture was stirred for 5 min. To facilitate the dissolution of sodium caseinate in water, the temperature was increased to 75 °C for 10 min. The solution was then cooled to 50 °C at which point, 0.3 g of KCl, 0.1 g of NaCl, and 0.01 g of NaN_3_, were added, and the mixture was stirred for 2 min. Subsequently, 0.2 g of CaCl_2_ were gradually added to this solution and stirred for another 5 min. The reaction was allowed to cool to room temperature (25 °C), enough DI water was added to maintain a volume of 100 mL and the solution was homogenized at 10,000 rpm for 2 min.

### 2.6. Preparation of NRTOCl and NRCl Oil-in-Water Emulsions for the Stability Study

Three different types of NRTOCl oil-in-water emulsions were made according to the following preparations.

### 2.7. Cas Emulsion

The emulsion was prepared using 480 mg of oil stock containing 15 wt% NRTOCl in canola oil and an aqueous phase (14.52 g) of Na–caseinate (2 wt%), KCl (0.3 wt%), NaCl (0.1 wt%), CaCl_2_ (0.2 wt%), and NaN_3_ (0.01 wt%). The oil phase was added to the aqueous phase and homogenized at room temperature at 16,800 rpm for 150 s.

#### 2.7.1. Cas–Lec Emulsion

The emulsion was prepared by using 480 mg of oil stock containing 15 wt% NRTOCl in canola oil and 20 mg of lecithin. The aqueous phase (14.50 g) of Na–caseinate (2 wt%), KCl (0.3 wt%), NaCl (0.1 wt%), CaCl_2_ (0.2 wt%), and NaN_3_ (0.01 wt%). The oil phase was added to the aqueous phase and homogenized at room temperature at 16,800 rpm for 150 s.

#### 2.7.2. Tween Emulsion

The emulsion was prepared by using 480 mg of oil stock containing 15 wt% NRTOCl in canola oil and an aqueous phase (14.52 g) of Tween 80 (2 wt%). The oil phase was added to the aqueous phase and homogenized at room temperature at 16,800 rpm for 150 s.

#### 2.7.3. NRCl Emulsion

The emulsion was prepared by using 480 mg of canola oil in an aqueous phase (14.52 g) of NRCl (19 mg), Na–caseinate (2 wt%), KCl (0.3 wt%), NaCl (0.1 wt%), CaCl_2_ (0.2 wt%) and NaN_3_ (0.01 wt%). The oil phase was added to the aqueous phase and homogenized at room temperature at 16,800 rpm for 150 s.

### 2.8. Sample Preparation of NRTOCl and NRCl Emulsions for Determination of Released Nicotinamide during the Stability Study

After each specific time of stability study for NRTOCl and NRCl emulsions, 1.5 mL of each emulsion was centrifuged (14,000 rpm) for 20 min at room temperature. Then, the aqueous phase was separated and filtered by a 0.22 μm filter for the determination of released nicotinamide by HPLC analysis.

### 2.9. Digestion Study

To show the bioavailability of NRTOCl by its release of NR, digestion of pure NRTOCl in simulated intestinal fluid as well as dissolved in MCT oil was investigated.

#### 2.9.1. Digestion Study of Pure NRTOCl Dispersed in Simulated Intestinal Fluid

The buffer for the simulated intestinal phase was prepared by using KCl, NaCl, KH_2_PO_4_, NaHCO_3_, MgCl_2_(H_2_O)_6,_ and HCl at pH 7 according to the Nature Protocol [30], with slight modifications. Specifically, in a 50 mL falcon tube, 60 mg (0.06 mmol) of NRTOCl were dissolved in 0.3 mL of ethanol and added to 10 mL of the buffer solution containing 400 mg bile bovine. To this solution, 0.75 mL of 0.3 M CaCl_2_ solution was added, and the pH was adjusted to 7 using 1 M HCl. Separately, 400 mg of fresh porcine pancreatin was dispersed in 4 mL of the buffer solution and then added to the NRTOCl mixture. The reaction flask was placed in an incubator at 37 °C with stirring at 250 rpm for 30 min at which time the pH of the sample was adjusted to 7 and incubation continued for an additional 30 min. This was repeated every 30 min for 2 h. To quench the enzymatic reaction, the flask was placed in an ice bath for 30 min, and then, the sample was centrifuged for 10 min at 14,000 rpm. The supernatant was removed and filtered using a 0.22 μm filter and analyzed for released NR and nicotinamide.

#### 2.9.2. Digestion Study of NRTOCl in MCT Oil in Simulated Intestinal Phase

For this procedure, 60 mg (0.06 mmol) of NRTOCl was dissolved in 150 mg of MCT oil to prepare 29 wt% NRTO in MCT oil. The aqueous phase was 10 mL of the buffer solution containing 400 mg bile bovine. The oil phase was added to the aqueous phase and homogenized at 15,000 rpm for 150 s at room temperature. Then, 0.75 mL of a CaCl_2_ solution (0.3 M) was added to this mixture, and the pH was adjusted around 7 by using HCl (1 M). Then, 400 mg of fresh porcine pancreatin was dispersed in 5 mL of buffer solution and added to the mixture. The sample was placed in an incubator (250 rpm) at 37 °C for 30 min. The reaction flask was placed in an incubator at 37 °C with stirring at 250 rpm for 30 min at which time the pH of the sample was adjusted to 7 using 1 M NaOH, and incubation continued for an additional 30 min. This was repeated every 30 min for 2 h. To quench the enzymatic reaction, the flask was placed in an ice bath for 30 min and then the sample was centrifuged for 10 min at 14,000 rpm. The supernatant was removed and filtered using a 0.22 μm filter and analyzed for released NR and nicotinamide.

## 3. Results and Discussion

We synthesized NRTOCl as a new compound using the reaction between NRCl and oleoyl chloride. The best result was obtained when the reaction was carried out in DMF as a solvent, and pyridine was used as the base. After purification of NRTOCl by column chromatography on silica gel, the pure product was characterized by FTIR, ^1^H NMR, ^13^C NMR, and LC–MS (details of the characterization including relevant spectra are contained in the Appendix A for this article Appendix A).

Although NRTOCl is a quaternary ammonium salt, it is not water soluble due to the three hydrophobic arms of the oleate ester. Therefore, to investigate the stability of NRTOCl with respect to hydrolysis, we dispersed NRTOCl in DI water by using ethanol as a cosolvent. A typical experiment was carried out by dissolving 15 mg of NRTOCl in 0.15 mL of ethanol and then adding 14.85 mL of DI water and gently shaking. The concentration of NRTOCl in this mixture was 1000 mg/L, the particle size and the zeta potential were 192 nm and +65 mV, respectively. Although the average size of NRTOCl in this sample was 192 nm, the TEM images showed smaller particles around 50 nm, with oval and spherical shapes with an aspect ratio close to unity (Figure 1). The NROTCl nanoparticles appear stacked on top of each other in layers. The highly positive charge of NRTOCl makes the nanoparticles stable in the aqueous phase.

After dispersing the ethanolic solution of NRTOCl in DI water, we dissolved NRCl in DI water and the stability of these samples was studied at 35 °C for 28 days (Figure 2). The concentration of this sample was 1000 mg/L.

We found that both NRTOCl and NRCl release nicotinamide during hydrolysis (Figure 3), and by measuring the released amount of nicotinamide in each sample by HPLC, we were able to calculate the remaining amounts of NRTOCl and NRCl. Release of NRCl from NRTOCl was not detected, which signifies that the oleate ester groups in NRTOCl are not hydrolyzed.

The results of the stability study indicate that NRTOCl is more stable than NRCl in DI water at 35 °C. The rate of hydrolysis of both compounds was similar for the first 4 days. After four days, however, the rate of hydrolysis of NRTOCl slowed significantly in comparison with NRCl. After 28 days, the remaining amount of NRTOCl is 53.3%, while the remaining amount of NRCl is 0.6% (Figure 2). By tri-functionalizing NRCl with long-chain ester groups, the hydrophobicity of NRTOCl increases, and consequently, the accessibility to the *N*-glycosidic bond by water decreases.

During the hydrolysis reaction, ribose–trioleate molecules, which are more hydrophobic than NRTOCl, are formed (Figure 4). This has the added benefit of protecting NRTOCl from hydrolysis because hydrolysis of the dispersed NRTOCl particles is carried out from the outer layers to the inner layers. As these layers are gradually converted to ribose trioleate they act as a superhydrophobic shell, which minimizes the penetration of water into the inner layer (Figure 4).

Since NRTOCl is a hydrophobic compound, to evaluate its stability in the oil phase, we created emulsions and studied the stability of NRTOCl as a function of time. For this purpose, first, we prepared a 15% *w/w* solution of NRTOCl in canola oil and used it to make three different emulsions using Na–caseinate, Tween 80, and lecithin (2 wt%) at room temperature. Initially, we found that Na–caseinate, and NRTOCl in DI water immediately formed aggregates. However, when the ionic strength of the aqueous phase was adjusted (NaCl (0.1 wt%), CaCl_2_ (0.2 wt%), and KCl (0.3 wt%)) Na–caseinate/NRTOCl emulsions were formed. The total oil phase in each emulsion was 3.2 wt% and the concentration of NRTOCl in the total volume of each emulsion (15 mL) was 4.4 mM. In the emulsions in which Na–caseinate was used as an emulsifier, we also added NaN_3_ (0.01 wt%) to prevent the growth of bacteria. When the NRTOCl emulsion was made by 2 wt% Na–caseinate, the average size and zeta potential in this emulsion (Cas emulsion) were 1020 nm and −14.4 mV, respectively. These results confirm that the droplets of NRTOCl in the oil phase are surrounded by caseinate anions. By using 2 wt% Na–caseinate and 2 wt% lecithins simultaneously as the emulsifiers, the average size and zeta potential of this NRTOCl emulsion (Cas–Lec emulsion) were 1012 nm and −13.3 mV. We also prepared an NRTOCl in canola oil-in-water emulsion by using 2 wt% Tween 80 as an emulsifier in DI water (Tween emulsion). The corresponding average size and zeta potential of this emulsion were 531 nm and +49.2 mV, respectively. As expected, because Tween 80 is overall neutral, the positive charge of NRTOCl droplets was almost intact in the emulsion.

After making the NRTOCl emulsions, we made NRCl in canola oil-in-water emulsion as a control experiment by using 3.2 wt% canola oil. The aqueous phase was a solution of Na–caseinate (2 wt%), NaCl (0.1 wt%), CaCl_2_ (0.2 wt%), KCl (0.3 wt%), and NaN_3_ (0.01 wt%). The concentration of NRCl in the total volume (15 mL) of this emulsion was 4.4 mM. We also prepared 15 mL of a 4.4 mM NRCl solution in DI water as the other control experiment.

We used the three NRTOCl emulsions and the two NRCl control samples to assess the stability of NRTOCl and NRCl to hydrolysis at 35 °C for 26 days (Figure 3a). The rate of degradation was determined by the amount of released nicotinamide from each sample. After 26 days, the percentage of NRTOCl that remained in the Cas, Cas–Lec, and Tween emulsions was 93.7, 90.3, and 80.0%, respectively. However, the percentage of NRCl that remained in NRCl emulsion and DI water was much lower, at 0.4% and 5.3%, respectively.

The results demonstrate that the stability of NRTOCl to hydrolysis in all three emulsions was significantly better than that of the NRCl either in emulsion or in water. Specifically, the stability of NRTOCl in the Cas emulsion was about 234 times more than NRCl in this emulsion system (93.7% NRTOCl:0.4%NRCl). Additionally, the Cas and Cas–Lec emulsions contained several anions and nucleophiles in the aqueous phase of the emulsion that increased the rate of NRCl hydrolysis. With respect to NRTOCl’s stability toward hydrolysis, Cas, and Cas–Lec emulsions showed better results than Tween emulsion. More than 90% NRTOCl remained intact in these samples throughout the 26-day period. This means that in comparison with Tween 80, Na–caseinate was able to stabilize NRTOCl droplets in the aqueous phase better.

As previously mentioned, caseinate anions with high surface activity surround the surface of NRTOCl droplets and neutralize the positive charge of NRTOCl on the outer surface. Moreover, due to the presence of calcium ions in the emulsion, calcium caseinate is formed and adsorbed onto the interface as a Pickering stabilizer [31], to make a thicker layer around the droplet and subsequently reduce the amount of water available to hydrolyze the NRTOCl. The other factor that can affect the stability of NRTOCl in these emulsions is the size of the droplets. For the Cas and Cas–Lec emulsions, the average size of the droplets was almost twice the average size of droplets in the Tween emulsion. This means that the surface areas of the droplets in the Cas and Cas–Lec emulsions were lower than those in the Tween 80 emulsion. Therefore, the accessibility of water to the NRTOCl droplets in the Cas and Cas–Lec emulsions was lower than that of the Tween 80 emulsion, and consequently, the rate of the NRTOCl hydrolysis in these emulations is lower than Tween emulsion.

As NRTOCl in the Cas and Cas–Lec emulsions showed good stability to hydrolysis at 35 °C, we studied the stability of NRTOCl in these emulsions at room temperature (25 °C) for an extended period of time (Figure 3b). In a new experiment, we found that after 42 days, the remaining amount of NRTO was 95.0% in Cas emulsion and 93.7% in Cas–Lec emulsion, while only 52.0% NRCl remained unhydrolyzed. Thus, the rate of NRTOCl hydrolysis in the emulsions, when compared with NRCl hydrolysis, was negligible. No phase separation of the emulsion was observed during the study period. However, there was a small increase in the average size of the NRTOCl droplets such that in the Cas emulsion, the average size increased from 1020 nm to 1281 nm, and in the Cas–Lec emulsion, the average droplet size increased from 1012 nm to 1106 nm. For all NRTOCl stability studies, no released NRCl was observed confirming that the oleate esters were stable to hydrolysis. Overall, NRTOCl in canola oil-in-water emulsions was more stable than dispersed NRTOCl and NRCl in water alone.

To prove that NRTOCl is digestible to NRCl, we assessed its digestibility in simulated intestinal fluid. For this enzymatic digestion, we used porcine pancreatin, bile bovine, and a buffer solution at pH 7 according to the Nature Protocol [30], with modifications. The lipase enzyme in porcine pancreatin can hydrolyze the ester groups to produce NRCl, nicotinamide riboside dioleate chloride (NRDOCl), nicotinamide riboside monooleate chloride (NRMOCl), and oleic acid, which are the main products of digestion (Figure 5). Nicotinamide (NAM) and ribose–trioleate (RTO) can also be formed, but they are a result of the hydrolysis of NRTOCl, not digestion.

After digestion, the released NRCl and nicotinamide were measured by LC–MS and HPLC. The results of SRM LC–MS show a single peak with 255.17 *m*/*z* that is in agreement with the structure of NR^+^ (Appendix A). In the mass spectrum, a fragment with 123.04 *m*/*z* is attributed to nicotinamide that formed by the elimination of ribose molecule from NRCl (Appendix A). After 2 h, 27.5% NRCl was released, which means that for this to occur, at least 27.5% NRTOCl was digested under the simulated intestinal phase conditions.

As a side reaction, 2.4% nicotinamide was observed in this reaction, which demonstrated that the hydrolysis of the *N*-glycosidic bond occurred during the NRTOCl digestion process, but this amount of hydrolysis was low, almost negligible. The other product of the NRTOCl hydrolysis was ribose–trioleate, which formed in roughly an equal amount as nicotinamide, 2.4%. However, this by-product was not dissolved in the aqueous phase and was measured indirectly. NRMOCl and NRDOCl are the other predicted products of NRTOCl digestion. We detected NRMO^+^ in the aqueous phase by LC–MS with 519.35 *m*/*z* but not NRDOCl. It seems that NRDOCl was not sufficiently dissolved in the aqueous phase to be detected. The main purpose of this study was to prove the digestibility and bioavailability of NRTOCl by its release of NRCl in the simulated intestinal phase. We believe that 27.5% of released NRCl is enough to prove the potential for NRTOCl to be a valuable ingredient for use in RTD applications.

Finally, we studied the digestion of NRTOCl in the oil phase. Compared with the pure digestion of NRTOCl, the amount of NRCl released was lower (11.3%). This result was expected because most of the oil phase was MCT oil, and consequently, the accessibility of lipase molecules to NRTOCl decreased. The product of *N*-glycosidic bond hydrolysis, nicotinamide, was detected during the digestion of NRTOCl in MCT oil. All of the obtained results confirmed the partial digestibility of NRTOCl (in both the pure form and in MTC oil) in a simulated intestinal phase.

## 4. Conclusions

In summary, we synthesized NRTOCl as a new hydrophobic NRCl derivative. The synthesis of NRTOCl was carried out by the reaction between NRCl and oleoyl chloride in the presence of pyridine. Pure NRTOCl was obtained in 64.3% yield and fully characterized. Due to the presence of three long-chain esters in the structure of NRTOCl, it was insoluble in water. Using EtOH as a cosolvent, NRTOCl could be dispersed in water as nanoparticles with an average size of 192 nm and a layered structure. We studied the stability of NRCl and NRTOCl in water at 35 °C for 28 days, and the results proved that NRTO was more than 88 times more stable than NRCl. Contrary to NRCl, NRTOCl was easily dissolved in canola, corn, and MCT oil at room temperature. This feature of NRTOCl helped us to evaluate its stability in the oil phase by making canola oil-in-water emulsions using sodium caseinate, lecithin, and Tween 80 as the emulsifiers. The stability of NRTOCl toward hydrolysis increased in canola oil-in-water emulsion by using sodium caseinate (2 wt%) as a food-grade emulsifier. In this system, after 26 days and at 35 °C, the amount remaining of NRTOCl and NRCl was 93.7% and 0.4%, respectively. These findings demonstrated that NRTOCl was about 234 times more stable than NRCl in this emulsion system. Additionally, we studied the stability to hydrolysis of NRTOCl in the canola oil-in-water emulsion system at room temperature (25 °C) for 42 days. These results of stability verified that the hydrolysis of NRTOCl was less than 5%, while 48% of the NRCl was hydrolyzed during this time. Finally, we observed that NRTOCl partially released NRCl in the presence of porcine pancreatin in a simulated intestinal phase. From the results of stability and digestibility experiments, we concluded that NRTOCl has the potential as an NRCl precursor in ready-to-drink (RTD) beverages and other food applications.

## Data Availability

The data presented in this study are openly available in FigShare at https://doi.org/10.6084/m9.figshare.17693816 (accessed on 18 November 2021).

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
