# Peer review of "Synthesis, Stability, and Bioavailability of Nicotinamide Riboside Trioleate Chloride"

_nutrients, 2021, doi:10.3390/nu14010113_

Round 1

Reviewer 1 Report

The manuscript "Synthesis, stability, and bioavailability of nicotinamide riboside trioleate chloride" by Zarei et al. reports the synthesis of a nicotinamide riboside (NR) derivative NRTOCl, which is more stable in water and oil-in-water emulsions compared to the currently used derivative NRCl. NRTOCl has the potential to be used as an NR derivative in beverages and other foods and supplements. NR is a more effective precursor of NAD, a coenzyme critical for several biological functions, compared to traditional precursors such as nicotinic acid or nicotinamide. 

The authors provide interesting data that clearly supports that NRTOCl is a stable NR derivative with the potential to be used in different beverage or foods. 

Minor comments/questions:

  • Figure 3 does has an incomplete color code and which bars are related to NRCl or NRTOCl, which makes it difficult to analyze.
  • NRTOCl digestion yields 27.5% and 11.3% in water and oil, respectively. How are these percentages compared to the release of  NR  by NRCl digestion? The stability assays were performed always in comparison to NRCl, but not the digestions and bioavailability assays. 

Reviewer 2 Report

This study reported new hydrophobic nicotinamide riboside (NR) derivative.

The results has the potential of practical application of new medicines and supplements..

However, introduction needs revision because there are several concerns as below.

In introduction (Line46-), authors described the relationship NAD+ and infectious disease, however, it has not been directly proven with respect to the effect on SARS-Cov2 infection in references 24-26. The description about COVID-19 and NAD was recently reported in “Heer CD, Sanderson DJ, Voth LS, Alhammad YMO, Schmidt MS, Trammell SAJ, Perlman S, Cohen MS, Fehr AR, Brenner C. Coronavirus infection and PARP expression dysregulate the NAD metabolome: An actionable component of innate immunity. J Biol Chem. 2020 Dec 25;295(52):17986-17996. doi: 10.1074/jbc.RA120.015138. Epub 2020 Oct 13. PMID: 33051211; PMCID: PMC7834058.”

  Furthermore, the effects of NR or NAD+ in vivo should not be overestimated considering main result of this study is new derivative not effects of NR. .
